# SEAT: Sparsified Enhancements for Attention Mechanisms in Time Series Transformers

## Abstract

Transformer models excel in time series tasks due to their attention mechanisms. However, they often suffer from "block-like" attention patterns caused by high feature correlation, leading to feature confusion and reduced performance. In this study, we mathematically prove and quantify this limitation, demonstrating how it affects the sparsity of the attention matrix and hinders effective feature representation. To overcome this issue, we propose a novel, model-agnostic, and plug-and-play method called SEAT (Sparsification-Enhanced Attention Transformer) that leverages frequency domain sparsification. By transforming time series data into the frequency domain, our method induces inherent sparsity, reduces feature similarity, and mitigates block-like attention, allowing the attention mechanism to focus more precisely on relevant features. Experiments on benchmark datasets demonstrate that our approach significantly enhances the accuracy and robustness of Transformer models while maintaining computational efficiency. This provides a mathematically grounded solution to inherent flaws in attention mechanisms, offering a versatile and effective approach for advancing time series analysis.

## 1 Introduction

In long-term sequence forecasting (LTSF) tasks, attention mechanisms play a crucial role in capturing dependencies within time series data. However, existing approaches, particularly those employing "Point-wise Attention" exhibit significant limitations. This strategy maps various features at each time step to embeddings, treating each as a token. Consequently, attention mechanisms allocate weights to each corresponding token. Experimental studies Zhang & Yan (2023) have revealed that, in conventional Transformers for LTSF tasks, cross-dimensional dependencies are not explicitly captured during the embedding process. This limitation adversely affects forecasting capabilities, as attention values tend to exhibit segmentation; adjacent data points often receive similar attention weights. The "Point-wise Attention" approach typically results in a block-like distribution of feature map values within time series representations. This block-shaped pattern causes approximate values of different features in the attention mechanism to become conflated, leading to model overfitting to noise. Consequently, the decision boundaries of the model become ambiguous, negatively impacting overall performance.

Subsequent research has attempted to mitigate these issues by employing alternative attention strategies, such as using pairs of patches. For instance, PatchTST Nie et al. (2023) utilizes patching and channel independence within Transformer architectures to significantly enhance performance, demonstrating that Transformers retain considerable potential for improvement in time series forecasting when appropriately adapted. Pathformer Chen et al. (2024) adopts a patch-based method to divide the time series into various temporal resolutions. Through this multi-scale division, dual attention is applied to these patches, enabling the capture of global correlations and local temporal dependencies.

Despite these advancements, existing methods primarily focus on enhancing feature extraction without fundamentally improving the quality of time series feature representation. Block-like attention may result in the attention weight matrix assigning significantly higher weights to certain time slices compared to others, leading to the aggregation of similar features within the attention output and increasing feature confusion. Such effects can adversely impact the model's learning capability and performance limits, as will be substantiated in the Methods section.

To address these challenges, we propose **SEAT**, a model-agnostic enhancement framework applicable to any Transformer architecture's input signals. SEAT employs frequency modelling across the entire time series and introduces a finite energy representation in the frequency domain, capitalizing on the sparse features present in time series data. This design enables the Transformer's attention mechanism to focus on independent feature representations as "Channel-wise Attention." By explicitly modelling the Fourier transform to reconstruct features as incremental signals, SEAT enhances the model's ability to distinguish approximate features and reduce feature redundancy and similarity. Consequently, this approach improves predictive performance in long-term forecasting tasks across various Transformer-based models. In summary, our contributions are threefold:

1. We provide a theoretical analysis of the limitations in attention mechanisms within time series forecasting Transformers, investigating the causes of feature confusion and susceptibility to overfitting.

2. Based on rigorous mathematical proofs, we design SEAT, a sparse sensing enhancement framework tailored for time series attention, ensuring the independence and sparsity of input features within Transformer architectures.

3. Our framework is decoupled from the underlying model architecture, offering plug-and-play functionality and compatibility with any existing Transformer-based architecture.

## 2 RELATED WORK

### 2.1 TIME-DOMAIN-BASED TRANSFORMERS

Time series forecasting has been significantly advanced by the integration of Transformer architectures Ashish (2017). Leveraging the self-attention mechanism, Transformer-based models have demonstrated exceptional performance in capturing long-range dependencies, a critical aspect for effective LTSF. Notable models in this domain include:

**Autoformer** Wu et al. (2021) introduced an auto-correlation mechanism specifically designed to leverage the inherent periodicity of time series data. This mechanism discovers dependencies and aggregates representations at the sub-series level, significantly improving the model's ability to utilize long-range information. By focusing on periodic patterns, Autoformer overcomes the information bottleneck that constrains the original Transformer architecture, thereby enhancing its forecasting performance for non-stationary time series.

**Pyraformer** Liu et al. (2022a) further addresses these limitations through a hierarchical time series decomposition approach. Its pyramid structure exhibits multiresolution properties, allowing the model to decompose time series into distinct temporal scales. At coarser resolutions, long-term dependencies are captured, while finer resolutions discern intricate short-term variations. This hierarchical decomposition facilitates a nuanced understanding of temporal dynamics, enhancing the model's ability to accurately forecast future trends and patterns. However, distinguishing long-term and short-term features is often limited by window parameter selection and data resolution, necessitating model structure modifications for different datasets, which reduces robustness and increases susceptibility to overfitting.

**Nonstationary Transformer** Liu et al. (2022b) introduces a dual approach to improve the modelling of non-stationary time series. It enhances data stationarity through techniques that mitigate non-stationarity within the time series and reformulates the Transformer's internal mechanisms to reintegrate non-stationary information. This twofold innovation significantly advances the Transformer's capability in handling non-stationary data, improving both predictability and forecasting performance.

**iTransformer** Liu et al. (2024) marks a significant advancement by surpassing traditional Transformer models in time series forecasting. It treats individual series as variate tokens, utilizing attention mechanisms to capture multivariate correlations and employing layer normalization and feed-forward networks to learn robust series representations. However, relying solely on time-domain tokens poses challenges in comprehensively portraying integral properties of time series data, such as overarching trends and periodic fluctuations. Additionally, the quadratic complexity and large parameter count inherent in Transformer models make them prone to overfitting, especially when applied to non-stationary time series.

## 2.2 Frequency-domain-based Transformers

Time-frequency transforms offer a promising avenue for transforming long-time series into sparse representations. The development of frequency domain methods can be traced back to the introduction of Fourier transforms and wavelet transforms, which provided novel perspectives for analyzing periodicity and global dependencies within time series data. Frequency-domain methods, traditionally used for signal processing tasks, excel at decomposing time series into their constituent frequencies, revealing periodic and seasonal components that are often crucial for accurate forecasting. Early frequency domain approaches, such as **FNet** Lee-Thorp et al. (2021) and **AFNO** Guibas et al. (2021), were primarily designed to enhance computational efficiency by leveraging Fourier transforms to replace self-attention or token-mixing mechanisms, thereby achieving faster computation.

There has been a growing trend towards integrating frequency-domain methods with sophisticated techniques such as attention mechanisms to achieve superior predictive performance in time series analysis. Traditional Transformer architectures applied to time series data rely on point-wise attention mechanisms, where individual temporal points undergo attention computations and predictions in isolation. As a result, these models often struggle to maintain and accurately model holistic, global features that inherently encode crucial information for accurate forecasting. In contrast, **FEDformer** Zhou et al. (2022) represents a paradigm shift by leveraging Transformer structures within the spectral domain for feature extraction. This innovative approach enables FEDformer to more effectively capture global characteristics vital for comprehending the intricate dynamics within time series data. By harnessing the complementary strengths of both temporal and spectral representations, FEDformer fosters a deeper, more nuanced understanding of underlying patterns and trends, thereby enhancing its predictive capabilities.

Similar to the objectives of numerous decomposition methods Wu et al. (2021); Zhou et al. (2021), the adoption of spectral domain approaches aims to facilitate the decomposition of time series data into distributions that are more conducive to learning. By transforming the time series into the frequency domain, these methods enable a more effective decomposition of temporal dynamics, transforming the data into representations that are easier for models to comprehend and utilize for prediction tasks. However, they share a common limitation: when dealing with complex time series, most frequency models tend to prioritize learning low-frequency features while overlooking high-frequency features, exhibiting a frequency bias. For example, **FITS** Xu et al. (2024) incorporates a low-pass filter in the frequency domain to capture essential time series information, but this inevitably results in the loss of high-frequency components. This bias can hinder the model's ability to fully capture the intricate dynamics present across all temporal scales, potentially impacting the accuracy and robustness of forecasts.

**Fredformer** Piao et al. (2024) aims to alleviate frequency bias by equally learning features across different frequency bands. This approach helps prevent the model from overlooking lower amplitude features crucial for accurate predictions. In these previous studies, attention layers are designed to function directly in the frequency domain to enhance spatial or frequency representations.

Transformer-based models for long-term time series forecasting (LTSF) have primarily advanced through two approaches. Time-domain models focus on enhancing attention mechanisms and employing patch-based training strategies. In contrast, frequency-domain models develop specialized filters and address biases between high and low-frequency components to improve performance. However, these methods mainly concentrate on updating and iterating Transformer architectures without deeply considering the inherent input properties of time series data and their impact on attention mechanisms, making it difficult to effectively mitigate high feature confusion. In the following sections, we provide a mathematical definition of this issue and introduce our **SEAT** framework to address it.

## 3 Method

### 3.1 Problem Statement

In the realm of time series forecasting (TSF), the problem statement can be formally defined as follows: Given a set of data points $\mathbf{X} = \{x_{t1}, \ldots, x_{tD}\}_{t=1}^{L} \in \mathbb{R}^{D \times L}$ within a lookback window of

a time series, where $L$ represents the size of the window, $D \geq 1$ is the number of variables, and $x_{tj}$ denotes the value of the $j$-th variable at the $t$-th time step. The objective of TSF is to predict the forecasting horizon $\hat{\mathbf{X}} = \{\hat{x}_{(L+1)1}, \ldots, \hat{x}_{(L+T)D}\}_{t=L+1}^{L+T} \in \mathbb{R}^{D \times T}$.

## 3.2 Solving Block-like Attention from the Input Side

The block-like distribution of attention feature maps is observed in many Transformer-based time series tasks, where the magnitudes of the feature values exhibit a block-like pattern. This phenomenon arises from the model's confusion among different approximate feature values. The conventional attention mechanism can be expressed by the following formula:

$$Attention(Q, K, V) = softmax\left(\frac{QK^T}{\sqrt{d_k}}\right)V \tag{1}$$

The phenomenon of "Block-like Attention" arises when contiguous elements within blocks of the attention score matrix exhibit similar magnitudes of values calculated by $A_{ij} = Q_i \cdot K_j^T$. This phenomenon signifies a clustering of high or low attention intensities within localized regions of the matrix, leading to a distinct block-like pattern. The presence of such patterns can be indicative of the model focusing its attention on specific subsets of input features or interactions, thereby providing insights into the model's decision-making processes. We can quantify the degree of feature confusion using the following mathematical definition, where the default method for $Similarity$ is the cosine similarity function and $f_i, f_j$ are features:

$$Sim(F) = \begin{cases} 0, & \text{if } N = 1 \\ \frac{1}{N*(N-1)} \sum_{i \neq j} Similarity(f_i, f_j), & \text{if } N \geq 2 \end{cases} \tag{2}$$

In the following, we define two primary attention mechanisms.

**Point-wise Attention (Temporal Attention) Definition:** Point-wise attention, also known as temporal attention, is a mechanism within deep learning architectures that assigns significance scores to individual data points or temporal instances within a sequence. This approach enables the model to dynamically adjust its focus on specific points in time, capturing nuances and salient features that may be crucial for downstream tasks. By selectively attending to these key points, point-wise attention enhances the model's ability to comprehend complex temporal patterns and dynamics within the data. Notably, the effectiveness of this approach has been demonstrated in seminal works (Ashish (2017); Hu et al. (2018); Nie et al. (2023)).

**Channel-wise Attention Definition:** Channel-wise attention, on the other hand, emphasizes the significance of individual feature channels within a multidimensional tensor. In this framework, each channel represents a unique feature map capturing distinct aspects of the input data. Channel-wise attention aims to dynamically reweight these feature channels, allowing the network to concentrate its representational power on the most informative and discriminative channels. Specifically, for time series data, each variable token is embedded into a high-dimensional space, where channel-wise attention operates to capture intricate multivariate correlations. By highlighting the channels that are most relevant to the task at hand, this mechanism enhances the overall feature representation, leading to improved performance in tasks such as classification, regression, or forecasting. Notable papers have highlighted the effectiveness of channel-wise attention (Wang et al. (2017); Woo et al. (2018); Liu et al. (2024)).

Compared to "Point-wise Attention" features, using multivariate data as tokens at the same time step results in naturally similar tokens for adjacent time steps. Although the sampling is discrete, the time series varies continuously, leading to many similar features that yield high similarity scores. Sparse feature representation facilitates the computation of a sparse attention matrix. This means that the attention matrix of size $N * N$ needs to meet a specified minimal error threshold $\epsilon > 0$, such that $N_0 = \sum_{i,j} \mathbb{I}(A_{ij} > \epsilon)$ is significantly less than $N^2$ where $\mathbb{I}(\cdot)$ is an indicator function that takes the value of 1 when the condition is true and 0 otherwise. Experiments show that sparse and independent feature sets yield a smaller $Sim$ value, indicating that sparse features exhibit lower similarity compared to dense features.

Standard Attention mechanisms exhibit certain limitations, notably the susceptibility to noise interference during weight computation, potentially leading to distractions on irrelevant elements. Beyond Top-k Attention, other sparse Attention mechanisms have emerged, such as Fixed Factorized

Attention and Strided Attention (Child et al. (2019)). According to our proposed metric Sim, models with lower feature confusion exhibit enhanced feature-capturing capabilities, fostering stronger generalization abilities. While sparse attention mechanisms theoretically risk overlooking certain global contextual information, they can partially mimic the effects of global context through meticulous design and optimisation. For instance, the BigBird (Zaheer et al. (2020)) adeptly captures long-range dependencies by blending sparse attention, global attention, and random skip connections.

However, in the domain of time series analysis, Block-like attention may hinder the model's ability to perceive global characteristics, while multi-scale attention frequently Chen et al. (2024) grapples with overfitting issues. Skip-connected attention, while improving attention focus, may compromise temporal resolution and lead to information loss, thereby restricting model performance. Prior research (Wu et al. (2020); Zhou et al. (2021); Nie et al. (2023)) has primarily focused on designing attention architectures and patterns. In contrast, our work demonstrates that incorporating sparsity at the input end can potentially address the shortcomings of existing attention mechanisms.

In our study, we propose **SEAT** to leverage frequency domain transformation to induce sparsity in time series data, thereby optimizing the performance of attention mechanisms from the input end. This approach is agnostic to the underlying attention model architecture, functioning as a versatile, plug-and-play component tailored specifically to enhance attention functionality. Our method offers a novel perspective on enhancing attention mechanisms, emphasizing the potential benefits of sparsity induction from the frequency domain at the input stage.

### 3.3 FOURIER DOMAIN OF THE TIME SERIES

The key to effectively addressing the challenge of block-wise attention lies in the ability of time-frequency transformation to impart sparse representations to time-series data. In this context, we will embark on a rigorous mathematical derivation to prove that long time-series data exhibit sparse representations under frequency domain transformations. Furthermore, we posit that by integrating feature partitioning into patches with frequency domain transformation, we can achieve an even more sparse representation of time-series data than either method alone.

**Theorem 1: Sparse Representation in the Frequency Domain** Given a signal $x(t)$ of time series data in $L^2$ space, if its Fourier transform $X(f)$ is supported on a finite set of frequencies, then $x(t)$ has a sparse representation in the frequency domain.

The Nyquist Sampling Theorem (Vaidyanathan (2001)), also known as the Shannon Sampling Theorem, fundamentally establishes the conditions under which a continuous-time signal can be accurately reconstructed from its discrete-time samples, ensuring that the Discrete Fourier Transform (DFT) yields equivalent results to the Continuous Fourier Transform (CFT) within the context of the sampled signal's representation. We assume that the sampled data of the time series satisfies the sampling theorem. This implies that the sampling frequency $f_s$ used to acquire the discrete samples $x[n]$ of the original continuous signal $x(t)$ is sufficiently high, specifically $f_s \geq 2B$, where $B$ is the maximum frequency content of the signal's spectrum $X(f)$.

Furthermore, we assume that the signals we acquire adhere to the finite bandwidth theorem, which states that the spectral content of the signal is confined within a limited frequency range. Given that the signals we collect are inherently discrete, due to the nature of the sampling process, it is crucial to consider the appropriate frequency resolution for the specific prediction task at hand. For instance, in tasks involving daily analysis, frequency features at the minute level may be deemed irrelevant or treated as noise. To ensure compliance with the finite bandwidth principle in practical sampling scenarios, we can adopt suitable sampling strategies that capture the signal within a reasonable frequency bandwidth range, thereby facilitating effective and efficient data processing for the intended prediction tasks.

**Proof of Theorem 1:** Given the assumptions mentioned above, we can deduce that in our long time-series analysis tasks, where the sampling theorem is satisfied, the use of the Discrete Fourier Transform (DFT) becomes equivalent to the Continuous Fourier Transform (CFT) for solving time-series problems. Thus, for these tasks, we can confidently employ the DFT as a valid and computationally efficient tool for analyzing and processing time-series data.

Assuming the signal $x[n]$ is expressed as $x[n] = \sum_{m=1}^{M} A_m e^{j\omega_m n}$ where $A_m$ represents the amplitude of the $m$-th frequency component, and $\omega_m$ denotes the angular frequency of the $m$-th frequency

component. It is further assumed that $\omega_m$ satisfies the condition $\omega_m = \frac{2\pi k_m}{N}$ where $k_m$ is an integer and $N$ is a constant that typically represents the total number of samples in the discrete domain.

The DFT of a discrete signal $x[n]$ is given by:

$$X[k] = \sum_{n=0}^{N-1} x[n]e^{-j\frac{2\pi}{N}kn} = \sum_{m=1}^{M} A_m \sum_{n=0}^{N-1} e^{j(\omega_m - \frac{2\pi}{N}k)}, \quad k = 0, 1, \ldots, N-1. \tag{3}$$

Under the frequency basis function we selected, the following equation can be calculated:

$$S_{m,k} = \sum_{n=0}^{N-1} e^{j(\omega_m - \frac{2\pi}{N}k)} = \begin{cases} N, & \text{if } k = k_m \\ \frac{1-e^{j2\pi(k_m-k)}}{1-e^{j\frac{2\pi}{N}(k_m-k)}} = 0, & \text{if } k \neq k_m \end{cases} \tag{4}$$

As a result, the frequency domain representation $X[k]$ exhibits non-zero values only at $k = k_m$, where $k_m$ corresponds to the indices of the frequency components present in the signal. Given that our time series can be arbitrarily long, the representation in the frequency domain of the "hidden space" of time series becomes sparse within frequency components.

### 3.4 FOURIER ATTENTION

Fourier attention operates by initially applying the Fourier Transform to the queries, keys, and values, subsequently conducting an attention mechanism within the frequency domain, and ultimately converting the outcomes back to the time domain via the inverse Fourier Transform. Let $F(\cdot)$ and $F^{-1}(\cdot)$ represent the Fourier Transform and inverse Fourier Transform, respectively. The Fourier attention mechanism can be formally expressed as follows:

$$Attention(Q, K, V) = F^{-1}\left(\sigma\left(\frac{F(Q)\overline{F(K)}^T}{\sqrt{d_q}}\right)F(V)\right), \tag{5}$$

where $\sigma(\cdot)$ denotes the softmax function, and $d_q$ is the dimensionality of the queries. Citing from previous research Zhang et al. (2022), calculating attention in the Fourier domain is equivalent to time-domain attention. Our method can be extended to attention in the frequency domain.

### 3.5 SEAT

Our proposed **SEAT** framework can be decomposed into several principal components, encompassing the Normalization Layer, SEAT Block, Feature Extraction Layer and Projection Layer.

**Normalization Layer:** Reversible instance normalization (RevIN)Kim et al. (2021), a generally applicable normalization-and-denormalization method with learnable affine transformation is well-known and widely used as the normalization layer. For better comparison of different models abilities, we use Revin normalization uniformly for all models.

**SEAT Block:** We drew a precise schematic diagram of the SEAT block structure in Fig 1. This novel framework is designed to induce sparsity in time series data through frequency domain transformations. This model-agnostic approach serves as a versatile and easily integrated component, compatible with any underlying attention architecture. By focusing on sparsity induction via frequency domain transformations, SEAT is specifically designed to enhance the overall performance of attention mechanisms from a new perspective in the input stage.

**Feature Extraction Layer:** This is a temporal feature extractor specifically based on transformer architecture since we designed SEAT to enhance the ac- curacy and robustness of attention mechanism. Two primary types of attention mechanisms can be introduced: one represented by "Channel-wise Attention" as in the case of iTransformer (Liu et al. (2024)), and the other by "Point-wise Attention" as exemplified by PatchTST (Nie et al. (2023)).

Regarding the whole processing pipeline of SEAT,as shown in Figure 1, we utilize the Revin normalization technique Kim et al. (2021) to preprocess the input time series. Subsequently, the sequence undergoes transposition and is fed into the SEAT block. Within this block, each time point of the individual series is embedded into variable tokens, facilitating the application of the Fast Fourier

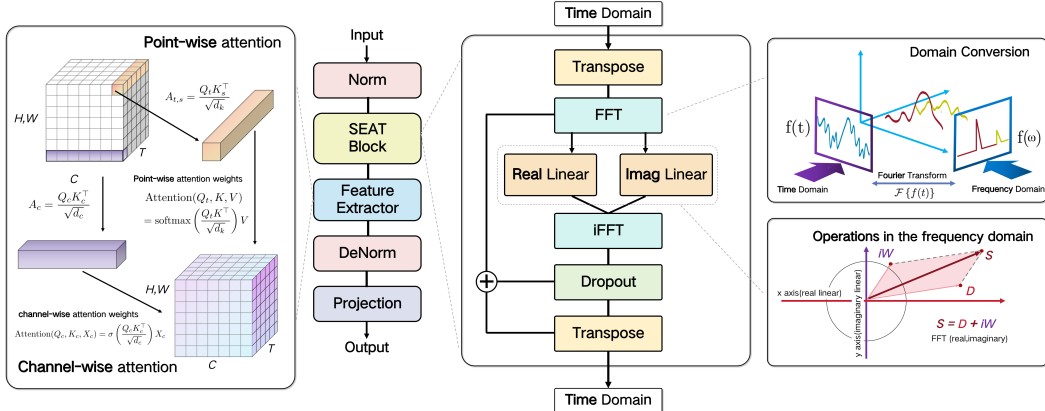

Figure 1: Overall Structure of SEAT.

Transform (FFT). This transformation converts the time series into the frequency domain. A linear module is then employed to enable the model to learn robust and sparse representations of the sequence. Following this, an Inverse Fast Fourier Transform (IFFT) and skip connection are applied to revert the sequence into the time domain. The resulting sparser representation of the sequence, once obtained, undergoes transposition and is subsequently input into an attention-based feature extractor. Ultimately, the sequence undergoes a de-normalization process and projection, yielding robust predictions for Long-Term Sequence Forecasting (LTSF).

In summary, while Channel-wise Attention focuses on the interrelationships between different feature channels, Point-wise Attention emphasizes dependencies between individual points or segments within the data. Both mechanisms are integral to enhancing model performance, depending on the specific structure and demands of the task at hand. Our SEAT model utilizes the iTransformer as the Temporal Feature Extractor, enabling it to serve as a versatile, plug-and-play component that can seamlessly integrate with any state-of-the-art (SOTA) transformer to enhance the model's overall performance.

## 4 EXPERIMENT

**Datasets:** we have undertaken extensive experiments on eight meticulously curated benchmarks, notably including the ETT datasets, which are subdivided into four distinct subsets: ETTh1, ETTh2, ETTm1, and ETTm2. Additionally, we have also employed the Weather, Exchange ECL, and Traffic datasets, adhering to the precedents established in Zhou et al. (2021); Zeng et al. (2023); Hebrail & Berard (2012); Zhao et al. (2019). These benchmarks, renowned for their rigour and comprehensiveness, provide a robust framework for assessing the performance and effectiveness of our forecasting models, particularly in the context of long-term horizon predictions.

**Baselines Compared:** Our proposed **SEAT** represents a model-agnostic approach that exhibits broad applicability to any deep neural network architecture. In this study, we extensively compare the well-acknowledged and advanced Transformers and designed a plug-and-play experiment to meticulously evaluate the efficacy of SEAT by integrating it into seven state-of-the-art Transformers designed specifically for time-series forecasting: iTransformer (Liu et al. (2024)), PatchTST (Nie et al. (2023)), Crossformer (Zhang & Yan (2023)), Pyraformer (Liu et al. (2022a)), Autoformer (Wu et al. (2021)), Informer (Zhou et al. (2021)), and Reformer (Kitaev et al. (2020)). This comprehensive experiment serves to validate the generality and enhancement capabilities of SEAT when applied to diverse yet sophisticated Transformer-based frameworks. The SEAT base model is typically employed with iTransformer as backbone architecture without explicit indication in the Main Result. We compute the MSE and MAE on Revin (Kim et al. (2021)) normalized data to measure different variables on the same scale. More details on experimental settings, including training details and hyperparameters, are provided in the Appendix. Experiments are implemented in PyTorch (Paszke et al. (2019)) and conducted on a single NVIDIA 4090 24G.

## 4.1 MAIN RESULT

Table 1: Mean result of **SEAT** versus other SOTAs transformers. In eight benchmark datasets, our method achieved first place in six out of the mean squared error (MSE) metrics and seven out of the mean absolute error (MAE) metrics.

| Models | SEAT Ours | | iTransformer 2024 | | PatchTST 2023 | | Crossformer 2023 | | Pyraformer 2022 | | Autoformer 2021 | | Informer 2021 | | Reformer 2020 | |
|---|---|---|---|---|---|---|---|---|---|---|---|---|---|---|---|---|
| Metric | mse | mae | mse | mae | mse | mae | mse | mae | mse | mae | mse | mae | mse | mae | mse | mae |
| ETTh1 | **0.436** | **0.433** | 0.454 | 0.447 | 0.469 | 0.454 | 0.529 | 0.522 | 0.865 | 0.731 | 0.518 | 0.500 | 1.078 | 0.813 | 0.961 | 0.757 |
| ETTh2 | **0.372** | **0.398** | 0.383 | 0.407 | 0.387 | 0.407 | 0.942 | 0.684 | 3.755 | 1.551 | 0.432 | 0.451 | 3.490 | 1.532 | 3.574 | 1.525 |
| ETTm1 | 0.396 | **0.400** | 0.407 | 0.410 | **0.387** | 0.400 | 0.513 | 0.496 | 0.750 | 0.615 | 0.583 | 0.513 | 0.948 | 0.717 | 0.928 | 0.688 |
| ETTm2 | **0.281** | **0.325** | 0.288 | 0.332 | 0.281 | 0.326 | 0.757 | 0.610 | 1.509 | 0.845 | 0.332 | 0.370 | 1.489 | 0.867 | 1.415 | 0.862 |
| Weather | **0.249** | **0.277** | 0.258 | 0.278 | 0.259 | 0.281 | 0.259 | 0.315 | 0.278 | 0.342 | 0.317 | 0.359 | 0.723 | 0.605 | 0.485 | 0.500 |
| ECL | **0.173** | **0.266** | 0.178 | 0.270 | 0.205 | 0.290 | 0.244 | 0.334 | 0.298 | 0.389 | 0.230 | 0.339 | 0.377 | 0.449 | 0.302 | 0.392 |
| Exchange | **0.349** | **0.402** | 0.360 | 0.403 | 0.367 | 0.404 | 0.940 | 0.707 | 1.308 | 0.945 | 0.493 | 0.493 | 1.411 | 0.968 | 1.000 | 0.837 |
| Traffic | 0.442 | 0.286 | 0.428 | 0.282 | **0.481** | **0.304** | 0.550 | 0.304 | 1.185 | 0.553 | 0.761 | 0.479 | 0.868 | 0.472 | 0.648 | 0.347 |
| 1st count | **6** | **7** | 1 | 1 | 1 | 0 | 0 | 0 | 0 | 0 | 0 | 0 | 0 | 0 | 0 | 0 |

**Main Results:** Table 1 presents concise forecasting outcomes, with the best accurate predictions highlighted in **red** and the second-best underlined. A lower MSE/MAE signifies superior prediction accuracy. **SEAT** ensures the independence and sparsity of input features within Transformer architectures. SEAT optimizes the utilization of input data, facilitating more efficient and effective processing within the attention layers of the Transformer model. SEAT method has demonstrated significant advantages across eight benchmark datasets. Our proposed model aggregates sparser features in the input end of the attention module and achieves the best result in long temporal modelling, enhancing the model's ability to handle high-dimensional time series and mitigating overfitting.

## 4.2 PLUG AND PLAY EXPERIMENTS

**Plug-and-play experiment:** In our plug-and-play experimental results, we observe that **SEAT** significantly improves the performance of various Transformer models. This phenomenon is clearly visualized in Figure 2. These visualizations demonstrate that SEAT effectively enhances prediction accuracy, regardless of whether it is applied to iTransformer, PatchTST, or other models. Table 2 illustrates the improvements in prediction accuracy by contrasting the original models with those augmented by SEAT. By calculating the percentage improvements, we clearly demonstrate the beneficial effects brought by this powerful plugin. As a model-agnostic enhancement, SEAT delivered immediate performance gains across all the models we evaluated. SEAT ensures the independence and sparsity of input features within Transformer architectures, optimizing the utilization of input data and facilitating more efficient and effective processing within the attention layers of the Transformer model. The substantial improvements observed across diverse datasets underscore its broad applicability and adaptability, thereby reinforcing SEAT's value as an effective tool for LSTF task.

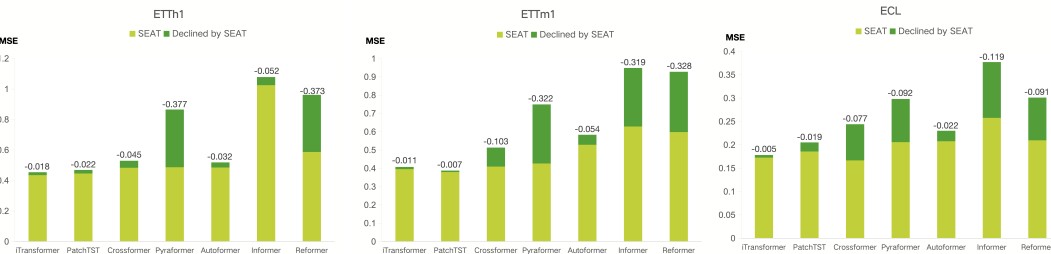

Figure 2: SEAT's Impact on Various Transformer Models' MSE Performance. The entire bars represent the original MSE scores, while the light green segments indicate the performance after applying the SEAT plugin, and the dark green segments represent the improvement specifically attributed to SEAT. These visualizations demonstrate the effectiveness of SEAT in enhancing the forecasting accuracy of diverse transformer models.

Table 2: Improvements of SEAT over different models with prediction lengths

| Models | | iTransformer | | PatchTST | | Crossformer | | Pyraformer | | Autoformer | | Informer | | Reformer | |
|---|---|---|---|---|---|---|---|---|---|---|---|---|---|---|---|
| | Metric | MSE | MAE | MSE | MAE | MSE | MAE | MSE | MAE | MSE | MAE | MSE | MAE | MSE | MAE |
| ETTh1 | Original | 0.454 | 0.447 | 0.469 | 0.454 | 0.529 | 0.522 | 0.865 | 0.731 | 0.518 | 0.500 | 1.078 | 0.813 | 0.961 | 0.757 |
| | +SEAT | 0.436 | 0.433 | 0.447 | 0.443 | 0.484 | 0.466 | 0.488 | 0.474 | 0.486 | 0.468 | 1.026 | 0.736 | 0.588 | 0.530 |
| | Improvement | +4.0% | +3.3% | +4.7% | 2.6% | +8.5% | +10.7% | +43.6% | +35.2% | +6.2% | +6.4% | +4.8% | +9.5% | +38.8% | +29.9% |
| ETTh2 | Original | 0.383 | 0.407 | 0.387 | 0.407 | 0.389 | 0.416 | 3.755 | 1.551 | 0.432 | 0.451 | 3.490 | 1.532 | 3.574 | 1.525 |
| | +SEAT | 0.372 | 0.398 | 0.374 | 0.402 | 0.394 | 0.416 | 0.433 | 0.434 | 0.459 | 0.449 | 0.677 | 0.571 | 0.459 | 0.451 |
| | Improvement | +2.9% | +2.2% | +3.4% | 1.2% | +58.2% | +39.1% | +88.5% | +72.0% | -6.3% | +0.4% | +80.6% | +62.7% | +87.2% | +70.4% |
| ETTm1 | Original | 0.407 | 0.410 | 0.387 | 0.400 | 0.513 | 0.496 | 0.750 | 0.615 | 0.583 | 0.513 | 0.948 | 0.717 | 0.928 | 0.688 |
| | +SEAT | 0.396 | 0.400 | 0.380 | 0.396 | 0.410 | 0.411 | 0.427 | 0.425 | 0.529 | 0.480 | 0.629 | 0.530 | 0.599 | 0.507 |
| | Improvement | +2.7% | +2.4% | +1.8% | 1.0% | +20.1% | +17.0% | +43.0% | +30.9% | +9.3% | +6.4% | +33.6% | +26.1% | +35.4% | +26.3% |
| ETTm2 | Original | 0.288 | 0.332 | 0.281 | 0.326 | 0.757 | 0.610 | 1.509 | 0.845 | 0.332 | 0.370 | 1.489 | 0.867 | 1.415 | 0.862 |
| | +SEAT | 0.281 | 0.325 | 0.279 | 0.326 | 0.292 | 0.332 | 0.302 | 0.336 | 0.304 | 0.342 | 0.389 | 0.406 | 0.318 | 0.348 |
| | Improvement | +2.4% | +2.1% | +0.7% | +0.2% | +61.4% | +45.7% | +80.0% | +60.2% | +8.4% | +7.6% | +73.9% | +53.2% | +77.5% | +59.6% |
| weather | Original | 0.258 | 0.278 | 0.259 | 0.281 | 0.259 | 0.315 | 0.278 | 0.342 | 0.317 | 0.359 | 0.723 | 0.605 | 0.485 | 0.500 |
| | +SEAT | 0.249 | 0.277 | 0.254 | 0.281 | 0.261 | 0.292 | 0.257 | 0.284 | 0.278 | 0.304 | 0.289 | 0.320 | 0.273 | 0.299 |
| | Improvement | +3.5% | +0.4% | +1.9% | -0.4% | -0.8% | +7.3% | +7.6% | +17.0% | +12.3% | +15.3% | +60.0% | +47.1% | +43.7% | +40.2% |
| ECL | Original | 0.178 | 0.270 | 0.205 | 0.290 | 0.244 | 0.334 | 0.298 | 0.389 | 0.230 | 0.339 | 0.377 | 0.449 | 0.302 | 0.392 |
| | +SEAT | 0.173 | 0.266 | 0.186 | 0.277 | 0.167 | 0.260 | 0.206 | 0.309 | 0.208 | 0.308 | 0.258 | 0.358 | 0.210 | 0.312 |
| | Improvement | +2.8% | +1.5% | +9.3% | 4.5% | +31.6% | +22.2% | +30.9% | +20.6% | +9.6% | +9.1% | +31.6% | +20.3% | +30.2% | +20.4% |
| Exchange | Original | 0.360 | 0.403 | 0.367 | 0.404 | 0.940 | 0.707 | 1.308 | 0.945 | 0.493 | 0.493 | 1.411 | 0.968 | 1.000 | 0.837 |
| | +SEAT | 0.349 | 0.402 | 0.365 | 0.405 | 0.367 | 0.414 | 0.396 | 0.429 | 0.459 | 0.466 | 0.368 | 0.428 | 0.448 | 0.459 |
| | Improvement | +3.1% | +0.2% | +0.3% | -0.2% | +61.0% | +41.4% | +69.7% | +54.6% | +6.9% | +5.5% | +73.9% | +55.8% | +55.2% | +45.2% |
| traffic | Original | 0.428 | 0.282 | 0.481 | 0.304 | 0.550 | 0.304 | 1.185 | 0.553 | 0.761 | 0.479 | 0.868 | 0.472 | 0.648 | 0.347 |
| | +SEAT | 0.442 | 0.286 | 0.477 | 0.291 | 0.479 | 0.311 | 0.794 | 0.436 | 0.713 | 0.392 | 1.030 | 0.567 | 0.638 | 0.333 |
| | Improvement | -3.3% | -1.4% | +0.8% | +4.3% | +12.9% | -2.3% | +33.0% | +21.16% | +6.2% | +18.2% | -18.7% | -20.1% | +1.4% | +4.0% |

## 4.3 ATTENTION STUDY

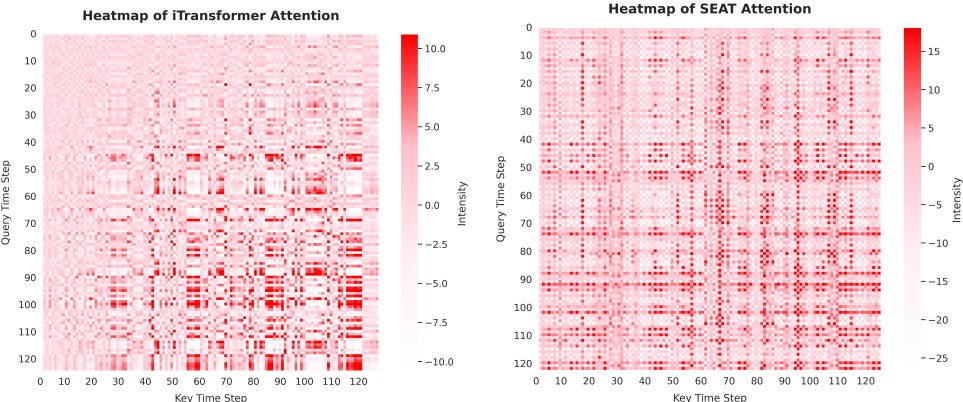

Figure 3: Self-attention scores from iTransformer and **SEAT** trained on ETTh1. The left heatmap represents the iTransformer's performance, which is likely the most advanced model to our knowledge. We integrate SEAT as a plugin into the iTransformer, and the right heatmap clearly illustrates the significantly improved self-attention scores achieved after applying SEAT. The visualization results demonstrate enhanced clarity and robustness in feature representation, thereby underscoring the significant efficacy of SEAT in augmenting the model's predictive capabilities.

Figure 3 presents a representative attention score map of the iTransformer for the Long-Term Short-Term Forecasting (LTSF) task. It is observable that the attention values in iTransformer tend to segment, with nearby data points sharing similar attention weights, resulting in ambiguous feature representations. Meanwhile, applying **SEAT** transforms the input signals into the frequency domain, leading to sparser and more distinct feature representations optimized for the attention mechanism. This transformation broadens the range of attention values and increases their distribution variance. The enhanced variance arises because frequency domain transformations decompose the time series into orthogonal frequency components, effectively reducing feature correlation and improving feature distinguishability. With sparser and less redundant features, the attention mechanism can assign a wider range of weights, more accurately reflecting the true importance of diverse features. A broader range and higher variance in attention values enable the model to differentiate more distinct and relevant features, thereby reducing feature redundancy and mitigating overfitting to noise.

Consequently, the attention mechanism in **SEAT** more effectively captures both cross-time and cross-dimension dependencies, resulting in clearer and more robust feature representations. These improvements facilitate more precise pattern recognition and enhance the overall learning capacity of the model, significantly boosting forecasting performance.

## 5 CONCLUSION

In this study, we introduce **SEAT**, a Sparse Sensing Enhancement Framework specifically designed to optimize attention mechanisms for time series forecasting within Transformer architectures. Through rigorous mathematical proofs and empirical analysis, we have demonstrated its feasibility and effectiveness. By ensuring the independence and sparsity of input features, the framework enhances both model performance and interpretability. Additionally, it overcomes the limitations of existing attention mechanisms by offering seamless plug-and-play functionality and compatibility with any Transformer-based architecture. This adaptability positions SEAT as a significant contribution to the field, paving the way for future advancements in attention-based models.

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

# A APPENDIX

## A.1 SAMPLING THEOREM

If the signal $x(t)$ possesses a limited bandwidth, with its spectrum $X(f) = 0$ for all $|f| > B$, then it can be perfectly reconstructed from samples taken at a sampling frequency $f_s$ that satisfies the condition $f_s \geq 2B$. The reconstruction is achieved using the following formula:

$$x(t) = \sum_{n=-\infty}^{\infty} x[n] \operatorname{sinc}\left(\frac{t - nT}{T}\right), \quad \text{where} \quad T = \frac{1}{f_s}. \tag{6}$$

In this equation, $x[n]$ represents the discrete samples of the signal taken at intervals of $T$, which is the reciprocal of the sampling frequency $f_s$. The sinc function, defined as $\operatorname{sinc}(x) = \frac{\sin(\pi x)}{\pi x}$, plays a crucial role in interpolating between the samples to reconstruct the continuous signal $x(t)$. This process is a direct consequence of the Nyquist-Shannon sampling theorem, which ensures that a bandlimited signal can be uniquely determined from its samples taken at a sufficiently high rate.

Therefore, we assume that the sampled data of the time series satisfies the sampling theorem. This implies that the sampling frequency $f_s$ used to acquire the discrete samples $x[n]$ of the original continuous signal $x(t)$ is sufficiently high, specifically $f_s \geq 2B$, where $B$ is the maximum frequency content of the signal's spectrum $X(f)$. Given this assumption, the discrete samples contain enough information to perfectly reconstruct the original continuous signal $x(t)$ using the reconstruction formula provided earlier.

## A.2 PATCHING AND SEAT MAKES SPARSER FEATURE REPRESENTATION FOR TIME SERIES

Consider a feature set $F$, where $F_s$ denotes a set of features $\{f_1, f_2, \ldots, f_N\}$. Each feature $f_i = (a_{i,1}, a_{i,2}, \ldots, a_{i,N})$ in $F$ is characterized by $N$ dimensions. To investigate the Patching algorithm through an inductive approach, we will initiate our discussion by focusing on the merging of a pair of features. We hypothesize that the $N^{th}$ and $(N-1)^{th}$ dimensions exhibit the highest degree of similarity. A single step in the patching process involves removing $f_N$, resulting in $f_p$, which represents one step of patching by eliminating highly correlated features.

Our objective is to evaluate whether this merging process, colloquially referred to as "Patching", leads to a sparser feature matrix by comparing the mathematical expressions before and after the operation. The new feature after patching can be defined as $\tilde{f}_i = (a_{i,1}, a_{i,2}, \ldots, a_{i,N})$ in $F_p$.

$$Sim(F) = \frac{\sum_{i \neq j} \sum_{k=1}^{N} a_{i,k} a_{j,k}}{N(N-1)} \tag{7}$$

$$Sim(F_p) = \frac{\left(\sum_{i \neq j, i, j \leq N-2} \sum_{k=1}^{N} a_{i,k} a_{j,k}\right) + 2 \sum_{k=1}^{N} \sum_{i=1}^{N-2} a_{i,k} a_{c,k}}{(N-1)(N-2)} \tag{8}$$

The assumption can be rewritten as follows:

$$\sum_{k=1}^{N} a_{N,k} a_{N-1,k} > \sum_{k=1}^{N} a_{i,k} a_{j,k}, \quad \text{for all } i \neq j \tag{9}$$

$$Sim(F) = \frac{\sum_{i \neq j, i, j \leq N-1} \sum_{k=1}^{N} a_{i,k} a_{j,k} + 2 \sum_{i < N} \sum_{k=1}^{N} a_{i,k} a_{N,k} + 2 \sum_{k=1}^{N} a_{N-1,k} a_{N,k}}{N * (N-1)} \tag{10}$$

Since the similarity between $f_N$ and $f_{N-1}$ is the largest, this leads to a significant contribution that is explicitly included in $Sim(F)$ but diminished in $Sim(F_p)$, therefore $Sim(F) \geq Sim(F_p)$. The Patching operation results in a sparser feature matrix.

The integration of feature segmentation into patches and frequency domain transformations has yielded a sparser representation of time series data compared to utilizing either method in isolation. Previous research (Nie et al. (2023); Du et al. (2023)) has employed patching techniques to divide univariate time series into patches, which can be either overlapped or non-overlapped. The key feature in PatchTST, characterized by channel independence and patching, can also be interpreted as feature independence and a sparser feature representation in our context.

## A.3 EXPERIMENT SETTING

we have standardized the parameters across all models to ensure a fair comparison on a uniform platform (time-series-library). Specifically, we have fixed the input dimension to 96 and varied the prediction horizon for time series forecasting, encompassing lengths of [96, 192, 336, 720]. The $batch\_size$ is set to 32, $learning\_rate$ is set to 1e-3, $d\_model$ is set to 512 and $dropout$ is set to 0.1.

Table 3: Performance comparison in terms of forecasting error metrics. This table illustrates the performance comparison among different models in terms of forecasting error metrics, adhering to a unified setting to ensure fairness. The mse and mae values highlight the accuracy of the predictions. The best-performing results are highlighted in **red**, while the second-best results are marked in blue with underlining. Lower MSE/MAE values signify higher predictive accuracy. The incorporation of SEAT into various benchmark attention models demonstrates significant performance enhancements, showcasing SEAT's effectiveness in improving the forecasting capabilities of Transformer-based models as a model-agnostic plugin.

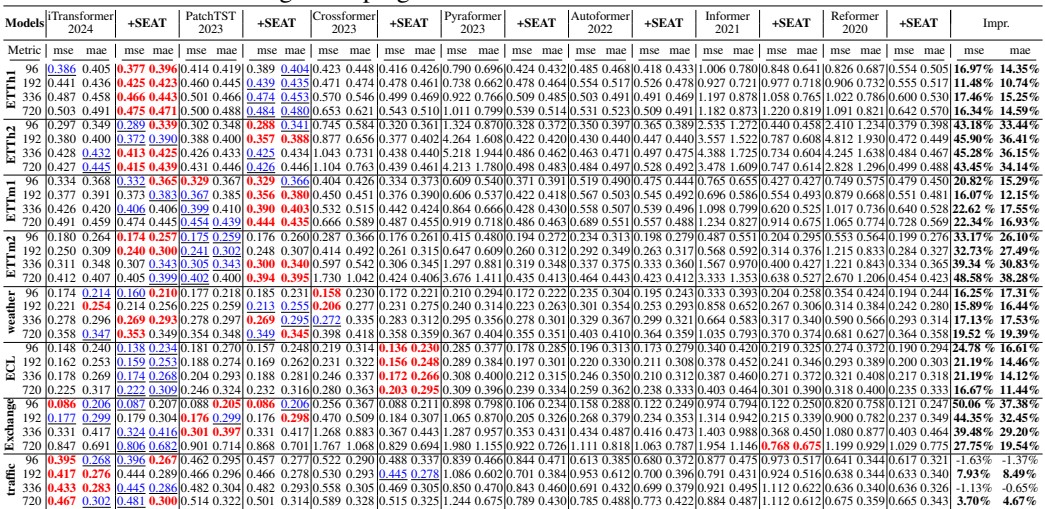

| Models | | iTransformer 2024 | | +SEAT | | PatchTST 2023 | | +SEAT | | Crossformer 2023 | | +SEAT | | Pyraformer 2023 | | +SEAT | | Autoformer 2022 | | +SEAT | | Informer 2021 | | +SEAT | | Reformer 2020 | | +SEAT | | Impr. | |
|---|---|---|---|---|---|---|---|---|---|---|---|---|---|---|---|---|---|---|---|---|---|---|---|---|---|---|---|---|---|---|
| Metric | | mse | mae | mse | mae | mse | mae | mse | mae | mse | mae | mse | mae | mse | mae | mse | mae | mse | mae | mse | mae | mse | mae | mse | mae | mse | mae | mse | mae | mse | mae |
| ETTh1 | 96 | 0.386 | 0.405 | 0.377 | 0.396 | 0.414 | 0.419 | 0.389 | 0.404 | 0.423 | 0.448 | 0.416 | 0.426 | 0.790 | 0.696 | 0.424 | 0.432 | 0.485 | 0.468 | 0.418 | 0.433 | 1.006 | 0.780 | 0.848 | 0.641 | 0.826 | 0.687 | 0.554 | 0.505 | 16.97% | 14.35% |
| | 192 | 0.441 | 0.436 | 0.425 | 0.423 | 0.460 | 0.445 | 0.439 | 0.435 | 0.471 | 0.474 | 0.478 | 0.461 | 0.738 | 0.662 | 0.478 | 0.464 | 0.554 | 0.517 | 0.526 | 0.478 | 0.927 | 0.721 | 0.977 | 0.718 | 0.906 | 0.732 | 0.555 | 0.517 | 11.48% | 10.74% |
| | 336 | 0.487 | 0.458 | 0.466 | 0.443 | 0.501 | 0.466 | 0.474 | 0.453 | 0.570 | 0.546 | 0.499 | 0.469 | 0.922 | 0.766 | 0.509 | 0.485 | 0.503 | 0.491 | 0.491 | 0.469 | 1.197 | 0.878 | 1.058 | 0.765 | 1.022 | 0.786 | 0.600 | 0.530 | 17.46% | 15.25% |
| | 720 | 0.503 | 0.491 | 0.475 | 0.471 | 0.500 | 0.488 | 0.484 | 0.480 | 0.653 | 0.621 | 0.543 | 0.510 | 1.011 | 0.799 | 0.539 | 0.514 | 0.531 | 0.523 | 0.509 | 0.491 | 1.182 | 0.873 | 1.220 | 0.819 | 1.091 | 0.821 | 0.642 | 0.570 | 16.34% | 14.59% |
| ETTh2 | 96 | 0.297 | 0.349 | 0.289 | 0.339 | 0.302 | 0.348 | 0.288 | 0.341 | 0.745 | 0.584 | 0.320 | 0.361 | 1.324 | 0.870 | 0.328 | 0.372 | 0.350 | 0.397 | 0.365 | 0.389 | 2.535 | 1.272 | 0.440 | 0.458 | 2.410 | 1.234 | 0.379 | 0.398 | 43.18% | 33.44% |
| | 192 | 0.380 | 0.400 | 0.372 | 0.390 | 0.388 | 0.400 | 0.357 | 0.388 | 0.877 | 0.656 | 0.377 | 0.402 | 4.264 | 1.608 | 0.422 | 0.420 | 0.430 | 0.440 | 0.447 | 0.440 | 3.557 | 1.522 | 0.787 | 0.608 | 4.812 | 1.930 | 0.472 | 0.449 | 45.90% | 36.41% |
| | 336 | 0.428 | 0.432 | 0.413 | 0.425 | 0.426 | 0.433 | 0.434 | 0.434 | 1.043 | 0.731 | 0.438 | 0.440 | 5.218 | 1.944 | 0.486 | 0.462 | 0.463 | 0.471 | 0.497 | 0.475 | 4.388 | 1.725 | 0.734 | 0.604 | 4.245 | 1.638 | 0.484 | 0.467 | 45.28% | 36.15% |
| | 720 | 0.427 | 0.445 | 0.415 | 0.439 | 0.431 | 0.446 | 0.426 | 0.446 | 1.104 | 0.763 | 0.439 | 0.461 | 4.213 | 1.780 | 0.498 | 0.483 | 0.484 | 0.497 | 0.528 | 0.492 | 3.478 | 1.609 | 0.747 | 0.614 | 2.828 | 1.296 | 0.499 | 0.488 | 43.45% | 34.14% |
| ETTm1 | 96 | 0.334 | 0.368 | 0.332 | 0.365 | 0.329 | 0.367 | 0.329 | 0.366 | 0.404 | 0.426 | 0.334 | 0.373 | 0.609 | 0.540 | 0.371 | 0.391 | 0.519 | 0.490 | 0.475 | 0.444 | 0.765 | 0.655 | 0.427 | 0.427 | 0.749 | 0.575 | 0.479 | 0.450 | 20.82% | 15.29% |
| | 192 | 0.377 | 0.391 | 0.373 | 0.383 | 0.367 | 0.385 | 0.356 | 0.380 | 0.450 | 0.451 | 0.376 | 0.390 | 0.606 | 0.537 | 0.422 | 0.418 | 0.567 | 0.503 | 0.545 | 0.492 | 0.696 | 0.586 | 0.554 | 0.493 | 0.879 | 0.668 | 0.551 | 0.481 | 16.07% | 12.15% |
| | 336 | 0.426 | 0.420 | 0.406 | 0.406 | 0.399 | 0.410 | 0.390 | 0.403 | 0.532 | 0.515 | 0.442 | 0.424 | 0.864 | 0.666 | 0.428 | 0.430 | 0.558 | 0.507 | 0.539 | 0.496 | 1.098 | 0.799 | 0.620 | 0.525 | 1.017 | 0.736 | 0.640 | 0.528 | 22.62% | 17.55% |
| | 720 | 0.491 | 0.459 | 0.474 | 0.445 | 0.454 | 0.439 | 0.444 | 0.435 | 0.666 | 0.589 | 0.487 | 0.455 | 0.919 | 0.718 | 0.486 | 0.463 | 0.689 | 0.551 | 0.557 | 0.488 | 1.234 | 0.827 | 0.914 | 0.675 | 1.065 | 0.774 | 0.728 | 0.569 | 22.34% | 16.93% |
| ETTm2 | 96 | 0.180 | 0.264 | 0.174 | 0.257 | 0.175 | 0.259 | 0.176 | 0.260 | 0.287 | 0.366 | 0.176 | 0.261 | 0.415 | 0.480 | 0.194 | 0.272 | 0.234 | 0.313 | 0.198 | 0.279 | 0.487 | 0.551 | 0.204 | 0.295 | 0.553 | 0.564 | 0.199 | 0.276 | 33.17% | 26.10% |
| | 192 | 0.250 | 0.309 | 0.240 | 0.300 | 0.241 | 0.302 | 0.248 | 0.307 | 0.414 | 0.492 | 0.261 | 0.315 | 0.647 | 0.609 | 0.260 | 0.312 | 0.292 | 0.349 | 0.263 | 0.317 | 0.568 | 0.592 | 0.314 | 0.376 | 1.215 | 0.833 | 0.284 | 0.327 | 32.73% | 27.49% |
| | 336 | 0.311 | 0.348 | 0.307 | 0.343 | 0.305 | 0.343 | 0.300 | 0.340 | 0.597 | 0.542 | 0.306 | 0.345 | 1.297 | 0.881 | 0.319 | 0.348 | 0.337 | 0.375 | 0.333 | 0.360 | 1.567 | 0.970 | 0.400 | 0.427 | 1.221 | 0.843 | 0.334 | 0.365 | 39.34% | 30.83% |
| | 720 | 0.412 | 0.407 | 0.405 | 0.399 | 0.402 | 0.400 | 0.394 | 0.395 | 1.730 | 1.042 | 0.424 | 0.406 | 3.676 | 1.411 | 0.435 | 0.413 | 0.464 | 0.443 | 0.423 | 0.412 | 3.333 | 1.353 | 0.638 | 0.527 | 2.670 | 1.206 | 0.454 | 0.423 | 48.58% | 38.28% |
| weather | 96 | 0.174 | 0.214 | 0.160 | 0.210 | 0.177 | 0.218 | 0.185 | 0.231 | 0.158 | 0.230 | 0.172 | 0.221 | 0.210 | 0.294 | 0.172 | 0.222 | 0.235 | 0.304 | 0.195 | 0.243 | 0.333 | 0.393 | 0.204 | 0.258 | 0.354 | 0.424 | 0.194 | 0.244 | 16.25% | 17.31% |
| | 192 | 0.221 | 0.254 | 0.214 | 0.256 | 0.225 | 0.259 | 0.213 | 0.255 | 0.206 | 0.277 | 0.231 | 0.275 | 0.240 | 0.314 | 0.223 | 0.263 | 0.301 | 0.354 | 0.253 | 0.293 | 0.858 | 0.652 | 0.267 | 0.306 | 0.314 | 0.384 | 0.242 | 0.280 | 15.89% | 16.44% |
| | 336 | 0.278 | 0.296 | 0.269 | 0.293 | 0.278 | 0.297 | 0.269 | 0.295 | 0.272 | 0.335 | 0.295 | 0.356 | 0.278 | 0.301 | | | 0.329 | 0.367 | 0.299 | 0.321 | 0.664 | 0.583 | 0.317 | 0.340 | 0.590 | 0.566 | 0.293 | 0.314 | 17.13% | 17.53% |
| | 720 | 0.358 | 0.347 | 0.353 | 0.349 | 0.354 | 0.348 | 0.349 | 0.345 | 0.398 | 0.418 | 0.358 | 0.359 | 0.367 | 0.404 | 0.355 | 0.351 | 0.403 | 0.410 | 0.364 | 0.359 | 1.035 | 0.793 | 0.370 | 0.374 | 0.681 | 0.627 | 0.364 | 0.358 | 19.52% | 19.39% |
| ECL | 96 | 0.148 | 0.240 | 0.138 | 0.234 | 0.181 | 0.270 | 0.157 | 0.248 | 0.219 | 0.314 | 0.136 | 0.230 | 0.285 | 0.377 | 0.178 | 0.285 | 0.196 | 0.313 | 0.173 | 0.279 | 0.340 | 0.420 | 0.219 | 0.325 | 0.274 | 0.372 | 0.190 | 0.294 | 24.78% | 16.61% |
| | 192 | 0.162 | 0.253 | 0.159 | 0.253 | 0.188 | 0.274 | 0.169 | 0.262 | 0.231 | 0.322 | 0.156 | 0.248 | 0.289 | 0.384 | 0.197 | 0.301 | 0.220 | 0.330 | 0.211 | 0.308 | 0.378 | 0.452 | 0.241 | 0.346 | 0.293 | 0.389 | 0.200 | 0.303 | 21.19% | 14.46% |
| | 336 | 0.178 | 0.269 | 0.174 | 0.268 | 0.204 | 0.293 | 0.188 | 0.281 | 0.246 | 0.337 | 0.172 | 0.266 | 0.308 | 0.400 | 0.212 | 0.315 | 0.246 | 0.350 | 0.210 | 0.312 | 0.387 | 0.460 | 0.271 | 0.372 | 0.321 | 0.408 | 0.217 | 0.318 | 21.19% | 14.12% |
| | 720 | 0.225 | 0.317 | 0.222 | 0.309 | 0.246 | 0.324 | 0.232 | 0.316 | 0.280 | 0.363 | 0.203 | 0.295 | 0.309 | 0.396 | 0.239 | 0.334 | 0.259 | 0.362 | 0.238 | 0.333 | 0.403 | 0.464 | 0.301 | 0.390 | 0.318 | 0.400 | 0.235 | 0.333 | 16.67% | 11.44% |
| Exchange | 96 | 0.086 | 0.206 | 0.087 | 0.207 | 0.088 | 0.205 | 0.086 | 0.206 | 0.256 | 0.367 | 0.088 | 0.211 | 0.898 | 0.798 | 0.106 | 0.234 | 0.158 | 0.288 | 0.122 | 0.249 | 0.974 | 0.794 | 0.122 | 0.250 | 0.820 | 0.758 | 0.121 | 0.247 | 50.06% | 37.38% |
| | 192 | 0.177 | 0.299 | 0.179 | 0.304 | 0.176 | 0.299 | 0.176 | 0.298 | 0.470 | 0.509 | 0.184 | 0.307 | 1.065 | 0.870 | 0.205 | 0.326 | 0.268 | 0.379 | 0.234 | 0.353 | 1.314 | 0.942 | 0.215 | 0.339 | 0.900 | 0.782 | 0.237 | 0.349 | 44.35% | 32.45% |
| | 336 | 0.331 | 0.417 | 0.324 | 0.416 | 0.301 | 0.397 | 0.331 | 0.417 | 1.268 | 0.883 | 0.367 | 0.443 | 1.287 | 0.957 | 0.353 | 0.431 | 0.434 | 0.487 | 0.416 | 0.473 | 1.403 | 0.988 | 0.368 | 0.450 | 1.080 | 0.877 | 0.403 | 0.464 | 39.48% | 29.20% |
| | 720 | 0.847 | 0.691 | 0.806 | 0.682 | 0.901 | 0.714 | 0.868 | 0.701 | 1.767 | 1.068 | 0.829 | 0.694 | 1.980 | 1.155 | 0.922 | 0.726 | 1.111 | 0.818 | 1.063 | 0.787 | 1.954 | 1.146 | 0.768 | 0.675 | 1.199 | 0.929 | 1.029 | 0.775 | 27.75% | 19.54% |
| traffic | 96 | 0.395 | 0.268 | 0.396 | 0.267 | 0.462 | 0.295 | 0.457 | 0.277 | 0.522 | 0.290 | 0.488 | 0.337 | 0.789 | 0.466 | | | 0.613 | 0.385 | 0.680 | 0.372 | 0.877 | 0.475 | 0.973 | 0.517 | 0.641 | 0.344 | 0.617 | 0.321 | -1.63% | -1.37% |
| | 192 | 0.417 | 0.276 | 0.444 | 0.289 | 0.466 | 0.296 | 0.466 | 0.278 | 0.530 | 0.293 | 0.445 | 0.278 | 1.086 | 0.602 | 0.701 | 0.384 | 0.953 | 0.612 | 0.700 | 0.396 | 0.791 | 0.431 | 0.924 | 0.516 | 0.638 | 0.344 | 0.633 | 0.340 | 7.93% | 8.49% |
| | 336 | 0.433 | 0.283 | 0.445 | 0.286 | 0.482 | 0.304 | 0.482 | 0.293 | 0.558 | 0.305 | 0.469 | 0.305 | 0.850 | 0.470 | 0.843 | 0.460 | 0.691 | 0.432 | 0.699 | 0.379 | 0.921 | 0.495 | 1.112 | 0.622 | 0.636 | 0.340 | 0.636 | 0.326 | -1.13% | -0.65% |
| | 720 | 0.467 | 0.302 | 0.481 | 0.300 | 0.514 | 0.322 | 0.501 | 0.314 | 0.589 | 0.328 | 0.515 | 0.325 | 1.244 | 0.675 | 0.789 | 0.430 | 0.785 | 0.488 | 0.773 | 0.422 | 0.884 | 0.487 | 1.112 | 0.612 | 0.675 | 0.359 | 0.665 | 0.343 | 3.70% | 4.67% |

## A.4 CODE OF ETHICS

We have read and understood the ICLR Code of Ethics, as outlined on the conference website. We fully acknowledge the importance of adhering to these ethical guidelines throughout all aspects of my participation in ICLR, including paper submission, reviewing, and discussions.

