# OpenReview forum: "SEAT: Sparsified Enhancements for Attention Mechanisms in Time Series Transformers"
_ICLR.cc/2025/Conference — ICLR 2025 Conference Withdrawn Submission_

### Official Review · Reviewer_WteU · 2024-10-27

**Soundness:** 3
**Presentation:** 3
**Contribution:** 3
**Rating:** 6
**Confidence:** 3

**Summary:**

This work introduces SEAT, a mechanism for mitigating block like attention patterns and potentially allowing the attention mechanism to focus on more relevant features. SEAT accomplishes this by first transforming the data to frequency domain. Experiments demonstrate the efficacy and potential of the proposed approach.

**Strengths:**

Originality :

The work is somewhat original. It uses known approaches in a novel manner as it uses frequency domain transformation as a mechanism to process the data into a form that mitigates block like patterns in  an attention mechanism for time series forecasting

Quality :

This work introduces a well structured framework that can be used as a plug-and-play module for time series forecasting. It is supported by experiments on multiple datasets that showcase the usefulness of the approach. However, there are some questions that come up.

Clarity :
This work is clear enough although there are some parts (such as those mentioned under Questions) that could be made more clear.

Significance:

Time series forecasting is an important line of exploration. The proposed approach introduces a plug-and-play module that can be incorporated in any state of the art approach to further enhance it and address some issues related to attention patterns. Overall, the proposed method is a step forward with potential applications beyond just time series forecasting tasks.

**Weaknesses:**

Although the work is well structured, it can benefit from deeper analysis of the proposed module and its influence on the different forecasting approaches along with comparisons with more varied baselines (as mentioned in the Questions section).

**Questions:**

So just to clarify, is it that the FFT is 2D FFT with the input $\in \mathbb{R}^{L \times D}$ ? The output of the SEAT block is $\in \mathbb{R}^{D \times L}$. Also,  in lines 341-347, it is not clear how the time domain (after IFFT and skip connection) representation is sparse. A frequency domain to time domain conversion need not necessarily produce a sparser output. How is it ensured that the output time domain representation will be sparse. Another question is where is the attention being applied in the SEAT block ? Is it in the Real Linear and Image Linear blocks ?


Does SEAT act as preprocessing step (that makes the time domain representation sparse) for the feature extractor which would then be used in the actual time series forecasting with standard (point/channel wise) attention ?


Table 1 claims significant advantages of SEAT over baselines. This is also demonstrated through Figure 2. However, several scores are very close to corresponding baselines when SEAT is applied (such as PatchTST for ETTh1 and ETTm1). Further more, there is a lot of variation in performance when SEAT is applied. Therefore, it would be great to have comparisons to show why some of the methods show more improvements than others or why the others do not show similar improvements (very small change as seen in PatchTST). There are even methods that show a drop in performance, such as Informer and iTransformer for the traffic set.


As the input and output of SEAT block are both time domain signals it would be beneficial to have a before and after comparison of these signals to analyze what was removed as part of the process.

In lines 316-317, it is mentioned that SEAT enhances accuracy and robustness. It would be great to have experiments that drive this point.

Some potential baseline that can be compared with SEAT are feature subset selection and feature disentanglement. These baselines can also help focus the attention mechanism on features that are important thereby potentially mitigating the block like attention pattern. The current experiments demonstrate that SEAT can be useful when added to an existing forecasting approach. However, it would also be useful to compare SEAT with other potential baselines such as those mentioned above.


minor questions :

Perhaps it would be useful to make it clear what $N$ and $F$ are in Equation 2. Is $F$ a subset of features from the window with size of $F$ being $N$ ?

How is $F$ chosen. This is in context of lines 211-212 where it is mentioned that a sparse attention mechanism yields a lower $Sim$ value. Isn't the value dependent of $F$. If so, then is the sparse attention mechanism providing a very different feature set? Also, it seems that lines 205-212 are related to channel wise attention. So is the sparse vs non-sparse comparison mentioned in this paragraph (and therefore $Sim$ calculation) based feature channels ?

---

### Official Review · Reviewer_oACe · 2024-10-29

**Soundness:** 2
**Presentation:** 1
**Contribution:** 1
**Rating:** 3
**Confidence:** 4

**Summary:**

The paper introduces SEAT (Sparsification-Enhanced Attention Transformer), a novel framework designed to improve Transformer models in time series forecasting by addressing "block-like" attention patterns that result in feature confusion and reduced performance. By applying frequency domain sparsification, SEAT reduces feature similarity and enables more precise focus on relevant data points, enhancing the robustness and accuracy of time series predictions. This model-agnostic, plug-and-play enhancement integrates with any Transformer architecture and was shown to outperform standard models across multiple benchmarks, thereby proving its effectiveness in boosting Transformer capabilities for long-term sequence forecasting.

**Strengths:**

1. The proposed method is simple and easy to understand.
2. SEAT is model-agnostic and operates as a plug-and-play solution.
3. Empirical results demonstrate that SEAT enhances the performance of the base Transformer method.

**Weaknesses:**

1. The main weakness is the insufficient evidence to support the claimed contribution. The author does not provide a rigorous analysis of the relationship between block-like attention and forecasting performance. The reason why SEAT can solve the problem is also not presented in detail. Consider providing quantitative metrics or visualizations that demonstrate how block-like attention patterns correlate with reduced performance. Additionally, the author could provide a more detailed explanation of SEAT's mechanism for addressing this issue, perhaps through step-by-step examples or comparative analyses with existing methods.

2. The proposed SEAT does not present enough novelty. The author should distinguish it from FITS [1], FreTS [2] and other methods using FFT. Consider providing a detailed comparison table or section that explicitly outlines the key differences between SEAT and other FFT-based methods like FITS and FreTS.

3. Too many typos and irregular expressions reduce the quality of the paper. For example, the theorem should be wrapped in a theorem environment. There should be no hyphens in the "accuracy" word in line 317.
4. The point-wise and channel-wise attention present different performances in previous research. They should not be analyzed separately.  The authors could conduct a comparative analysis that shows how these attention mechanisms interact and jointly influence model performance.


[1] Zhijian Xu, Ailing Zeng, Qiang Xu: FITS: Modeling Time Series with 10k Parameters. ICLR 2024
[2] Kun Yi, Qi Zhang, Wei Fan, Shoujin Wang, Pengyang Wang, Hui He, Ning An, Defu Lian, Longbing Cao, Zhendong Niu:
Frequency-domain MLPs are More Effective Learners in Time Series Forecasting. NeurIPS 2023

**Questions:**

See weaknesses.

---

### Official Review · Reviewer_Pa7y · 2024-11-01

**Soundness:** 3
**Presentation:** 2
**Contribution:** 2
**Rating:** 5
**Confidence:** 4

**Summary:**

This paper proposes SEAT, which employs frequency domain representations to sparsify input signals and mitigate certain issues that hinder time series Transformers such as block-like attention. SEAT shows competitive empirical performance and can be applied across the board to improve model performance.

**Strengths:**

I think the idea of employing frequency-based representations for time-series is very good, and is a trend in the literature I appreciate. The authors did a good job in identifying the need for sparsification for the Transformer backbone to operate in a performant manner and the empirical results are also strong.

**Weaknesses:**

- There are a few typos throughout the text that the authors should look out for and correct: I spotted some in L291 F() and F^-1() and in L317  acc-uracy.
- I believe that the authors should tone down some of the wording around their theoretical contributions. I would not consider the use of the Nyquist theorem " embarking on a rigorous mathematical derivation"
- I think that the authors should highlight the computational cost of their method and how it compares to other models.
- An ablation showing the effect of dealing with block-like attention and adding channel-wise attention would be useful to see what is driving the performance of the model
- While the authors did a good job in citing related work, I think they should have a dedicated paragraph in 2.1 to RFormer [1], which also sparsifies the input stream and learns cross-dependencies between time series through the signature transform.

I am willing to raise my score if the authors address these concerns

[1] Moreno-Pino, Fernando, et al. "Rough Transformers: Lightweight Continuous-Time Sequence Modelling with Path Signatures." arXiv preprint arXiv:2405.20799 (2024).

**Questions:**

- Could the authors elaborate on the computational benefits/tradeoffs of SEAT?

---

### Official Review · Reviewer_8zJ5 · 2024-11-02

**Soundness:** 2
**Presentation:** 2
**Contribution:** 2
**Rating:** 3
**Confidence:** 3

**Summary:**

This paper introduced SEAT(Sparisification-Enhanced Attention Transformer) to address time-series forecasting ability of transformers. SEAT derives sparsity of attention mechanism by transforming to frequency domain through Fourier transform, thereby solving problem of channel confusion present in original attention. One advantage of this method is that it is plug-and-play type, which can be applied to any kind of transformer-based models. The contributions are 1) analysis on mathematical limitation on attention mechanism, 2) enhancement of feature independence through sparsity, and 3) plug-and-play functionality compatible  with various transformers.

**Strengths:**

1. The paper provides a mathematical analysis of the limitations in existing attention mechanisms, offering motivation for SEAT’s design and highlighting its theoretical foundation.
2. SEAT functions as a model-agnostic, plug-and-play enhancement applied at the transformer’s input stage, making it highly practical and easy to integrate with various architectures.
3. SEAT consistently improves performance, demonstrating substantial reductions in MSE and MAE across benchmark models, affirming its effectiveness over baseline approaches.

**Weaknesses:**

1. The reliance on frequency transformation and inverse transformation introduces additional computation steps in SEAT, which may lead to increased model complexity. This added computational burden could impact the practical efficiency of SEAT in large-scale applications, particularly when low-latency processing is essential.
2. Although the authors endeavor to mathematically validate SEAT’s theoretical soundness by establishing two initial assumptions, there remains ambiguity regarding the generalizability of these assumptions across different time series contexts. Without explicit discussion of potential limitations, the universality of these assumptions might be overestimated.
3. The sparsity-inducing approach SEAT employs could vary in effectiveness depending on the characteristics of the time series data. Therefore, further experimentation is required to determine whether SEAT’s sparsification strategy can yield consistent benefits across diverse types of time series data, particularly in scenarios with complex temporal dynamics or non-stationary patterns.

**Questions:**

1. Considering SEAT’s additional computations, is its efficiency in forecasting performance still competitive? A detailed analysis of its computational complexity and processing speed relative to conventional methods would help clarify its practicality for real-time applications.
2. The attention map presented for the ETTh1 dataset seems to effectively address block-wise attention issues. Do other datasets exhibit similar attention patterns? Expanding the attention map analysis across various datasets would provide insights into whether SEAT consistently enhances feature focus or if its benefits are dataset-specific.

---

### Official Review · Reviewer_NbTn · 2024-11-03

**Soundness:** 2
**Presentation:** 2
**Contribution:** 2
**Rating:** 3
**Confidence:** 4

**Summary:**

The paper introduces SEAT (Sparsified Enhancements for Attention Mechanisms in Time Series Transformers), a framework designed to improve the performance of Transformers in time series forecasting. Transformer models, while effective, often suffer from block-like attention patterns due to high feature similarity, which reduces their ability to accurately capture complex dependencies in time series data. SEAT addresses this limitation by transforming input signals into the frequency domain, which introduces sparsity and reduces feature similarity. As a result, SEAT enhances the Transformer’s ability to focus on relevant features, improving accuracy and robustness in long-term forecasting tasks.

**Strengths:**

- The authors address why previous Transformer-based approaches may not perform optimally for long-term time series forecasting, highlighting that redundancy in attention mechanisms can hinder model performance, especially when similar patterns recur over extended periods.
- SEAT is designed to be adaptable across various Transformer-based architectures by substituting their attention mechanisms. It is straightforward to implement, demonstrating promising potential with minimal effort.

**Weaknesses:**

- While the authors describe their derivations as a “rigorous mathematical analysis,” the content could be simplified and more concisely explained by referencing established textbooks on Fourier analysis [1]. The authors may benefit from adopting a more modest tone in their presentation.
[1] Elias M. Stein, Fourier Analysis: An Introduction.

- There is a lack of detailed explanations for the experimental results, particularly regarding why SEAT outperforms baseline models on certain datasets but underperforms on others.

**Questions:**

- Can you explain how much parameters were increased when SEAT applied? It's unclear whether the performance gains may result from an increase in parameters.
- Can you clarify where Fourier attention is applied, perhaps by including an annotated figure? Figure 1 and its explanation currently lack sufficient detail on this aspect.
- Althogh the author explains their usages of fourier transform, the results might be comprehended in different way. According to Thm 1, the time series seems to be discretized and sparsified, in the perspective of transformer, the time(or frequency) interval of adjacent samples is not important. In Transformers, relative position matters more than absolute time points. Rather than that, the performance improvements might be due to the fact that as the time-seires get longer, the patterns of time series becomes regular, which means time series on frequency domain are clusterd, easily captured by transformer. Could the authors provide more insight on this perspective?
- Could you explain under which conditions SEAT consistently outperforms or underperforms compared to other methods across different datasets?

---

### Official Review · Reviewer_DJqv · 2024-11-04

**Soundness:** 2
**Presentation:** 2
**Contribution:** 2
**Rating:** 3
**Confidence:** 3

**Summary:**

This paper analyzes the limitations of existing "block-like" attention patterns in time-series forecasting models. To address this challenge, it proposes a model-agnostic method to introduce sparsity by leveraging frequency-domain transformations. Experiments across various Transformer time-series forecasting models demonstrate the effectiveness of the proposed method.

**Strengths:**

1. The paper proposes to introduce sparsity to address the "block-like" attention challenge through frequency-domain transformations.

2. The proposed approach is model-agnostic and has shown improvement across various Transformer-based time-series forecasting models.

**Weaknesses:**

1. Based on the descriptions in Section 3.5, the SEAT block applies FFT, a linear transformation in the frequency space, followed by IFFT to convert the time series to time domain. How does this process specifically introduce sparse representation? Moreover, the attention mechanism still takes place in the time domain. Why does Section 3.4 introduce Fourier attention?

2. How does the proposed method compare with FEDformer which applies random masking in the frequency domain?

3. In line 298, the statement, "Citing from previous research Zhang et al. (2022), calculating attention in the Fourier domain is equivalent to time-domain attention", is only valid in the linear case without the softmax operation, as mentioned in the original paper.

4. The Related Work section provides too many details on individual papers instead of offering a summarized overview.

5. In line 317, there is a typo of "ac- curacy".

**Questions:**

1. Can SEAT improve the performance of current LLM-enhanced forecasting models such as GPT4TS [1], S2IP-LLM [2], and Time-LLM [3]?

2. Can SEAT improve non-Transformer models such as DLinear [4]?

[1] One Fits All: Power General Time Series Analysis by Pretrained LM

[2] S2IP-LLM: Semantic Space Informed Prompt Learning with LLM for Time Series Forecasting

[3] Time-LLM: Time Series Forecasting by Reprogramming Large Language Models

[4] Are Transformers Effective for Time Series Forecasting?

---

### Note · Authors · 2024-11-19

I have read and agree with the venue's withdrawal policy on behalf of myself and my co-authors.